# AI Model Modulation with Logits Redistribution

## Abstract

The substantial data and resource consumption of training deep neural networks has rendered the large-scale training accessible only to organizations with necessary infrastructure and massive datasets. Once these models are developed, they are typically adapted to meet the diverse requirements of model owners and users through techniques like early exiting and fine-tuning. However, maintaining multiple specialized versions of the established model is inefficient and unsustainable in the long run. In response to this challenge, we propose AIM, a novel *model modulation* paradigm that enables a single model to exhibit diverse behaviors meeting the specific needs of stakeholders. AIM enables two key modulation modes: *utility* and *focus* modulation. The former provides model owners with dynamic control over output quality to deliver varying utility levels from the same model, the latter offers users precise control to shift model's focused features of inputs.

AIM introduces a *logits redistribution strategy* for modulating model behaviors. It operates in a *training data-agnostic* and *retraining-free* manner by directly manipulating off-the-shelf pre-trained networks, facilitating AIM's seamless integration across diverse neural network architectures. To mathematically guarantee that our modulation achieves a precise regulation of model behavior, we establish a formal foundation grounded in the statistical properties of logits ordering via joint probability distributions. Our evaluation spans across diverse applications, including image classification, semantic segmentation, and text generation, utilizing prevalent architectures such as ResNet, SegFormer, and Llama. Experimental results confirm the efficacy of our approach, demonstrating the practicality and versatility of AIM in realizing AI model modulation. AIM provides both theoretical and system-level tools to empower a single model to meet diverse needs of both model owners and users, paving the way for scalable, accessible, and efficient AI deployment.

## 1 Introduction

Deep neural networks (DNNs) have revolutionized various industries such as healthcare [17], finance [5], autonomous vehicles [30], and natural language processing [21], enabling significant breakthroughs in tasks like image recognition [20], semantic segmentation [26], and language translation [38]. Despite their success, the development of high-quality models demands extensive computational resources, massive datasets, and substantial financial investment. This has restricted large-scale training to organizations with necessary infrastructure, as seen with GPT-3 [3], which comprises 175 billion parameters and takes 355 GPU-years and $4.6M$ for a single training run [6, 34].

While the AI community continues to push the boundary of model performance in complex tasks, a critical challenge in the new era of AI revolves around managing the intellectual property of established models and adapting them to meet diverse needs of downstream tasks. Specifically, for model owners, the ability to maintain *controllability* is paramount, which enables them to deploy and customize models for different market segments and operational environments with varying business goals. For model users, they seek *adaptability*, desiring models that can adjust their behavior to suit individual preferences and contextual needs. These demands are illustrated by two typical scenarios presented below:

***Scenario #1 (model owners)***. An online service provider offers different service tiers. Free-tier users receive lower-quality outputs, such as reduced resolution or basic code suggestions. Premium users, however, get enhanced results with higher quality and additional features. Real-world examples include cutout.pro [1] and together.ai [2], which provide models with free low utility options or varying capabilities at different price points.

***Scenario #2 (users)***. Individual users interacting with AI systems, such as driving assistance platforms, often seek adaptability in the model's behavior to suit their preferences [11, 13, 29]. For instance, one driver may prioritize highlighting vehicles on the road, while another may emphasize detecting pedestrians. Such personalization has been offered in advanced driver assistance systems (ADAS) [11] to match individual driving styles like assertive or defensive driving, which improves user comfort and acceptance [13, 29].

Traditional techniques such as early exit [23, 35, 41, 44] and fine-tuning [16, 28] can be employed to control model utility or adapt established models to specific tasks or constraints. Early exit introduces intermediate exit points at different layers within a neural network, allowing early termination of inference for faster but potentially less accurate predictions. However, implementing early exit requires architectural modifications, which may not always be feasible due to limited model accessibility and can complicate integration and maintenance. Fine-tuning adjusts a pre-trained model to a new task by retraining it on a smaller, task-specific dataset. Nevertheless, fine-tuning requires access to training data and involves retraining or additional optimization steps [4, 16, 25, 28]. Even though both techniques can produce multiple tailored versions, managing these versions across a large user base is impractical, as the cost of maintaining consistency and applying updates across versions is prohibitively high. These limitations underscore the need for a flexible, lightweight approach that allows modulation of the model's usage without retraining or altering the model's architecture.

**Our work**. In this work, we propose a novel paradigm of *model modulation* that enables a *single* model to exhibit diverse behaviors, so as to satisfy the requirements of different utility levels or different feature focuses with a single model. This paradigm holds broad applicability in modern AI deployment, where controlling model utility levels or adjusting model prioritization is critical, such as in machine learning as a service (MLaaS) [31] and on-device deployment [27]. Ideally, the model modulation gets rid of the necessity of altering the underlying model parameters or architecture, and introduces controlled adjustments to the model's responses. The core research question of model modulation lies in *how to dynamically adapt the performance and behavior of a single model without the burden of retraining or maintaining multiple separate versions.*

We introduce A​ɪ​ᴍ (AI Modulator) as an approach to model modulation. It supports two types of modulation modes: *utility* and *focus* modulation. Utility modulation makes the model output deviate from the original output, which is useful in scenarios where restricted responses are desired *(Scenario #1)*. Focus modulation aims to make the model more responsive to specific areas of interest, which is helpful for subsystems of safety-critical systems to anticipate specific potential hazards *(Scenario #2)*. The challenges to address by A​ɪ​ᴍ are at least twofold. For model owners seeking *controllability*, it is important to ensure that utility modulation preserves the model's core knowledge so that, even when output quality is reduced, the outputs remain meaningful (*e.g.,* large language models should always deliver coherent outputs across utility levels) and do not compromise the model's integrity *(Challenge #1)*. For users desiring *adaptability*, balancing the trade-off in focus modulation between emphasizing specific inputs (such as prioritizing certain features in ADAS) and maintaining overall performance is essential, as too much intervention would affect the model's effectiveness in other areas *(Challenge #2)*.

To maintain the model's core knowledge *(Challenge #1)*, A​ɪ​ᴍ avoids altering feature-learning structures within the model. Instead, it directly operates on and strategically adjusts the model's logits to transform the original network (denoted as $f^*$) into a modulated network (denoted as $f^\epsilon$) that exhibits target behaviors. In particular, it incorporates a *control function* $\phi$ that redistributes the model's logits by adjusting their values according to specific probability distributions. This allows for fine-grained control while maintaining the model's integrity. Besides enabling model owners to offer varying utility tiers, this granular approach strikes a balance between responsiveness to specific features and overall performance. This flexibility allows users to tailor the model's behavior to their needs, enhancing responsiveness without compromising the model's overall effectiveness *(Challenge #2)*. Since logits serve as a common intermediate representation across architectures, A​ɪ​ᴍ operates as a *training data-agnostic* and *retraining-free* process by directly modifying off-the-shelf trained networks, making it well-suited for seamless integration across diverse neural network architectures.

We provide a robust formal foundation as the theoretical guarantee of A​ɪ​ᴍ's effectiveness. Its core is to establish a direct relationship between the model's behavior pre- and post- logits redistribution. By analyzing the statistical properties of logits through joint probability distributions, we quantify how controlled interventions affect their distribution and order. Our formal analysis ensures that, given specific conditions on the logits' distribution, the probability of achieving a desired modulation outcome can be precisely controlled. This formalization lays the groundwork for a probabilistic analysis of model behavior, offering a solid formal foundation for AI model modulation.

We conduct extensive evaluations across a wide range of application domains and model architectures to validate A​ɪ​ᴍ. Our evaluation spans image classification, semantic segmentation, and text generation, utilizing prevalent deep neural network architectures such as ResNet-56 [14], SegFormer-B2 [42], and Llama-3.1-8B [37]. Through utility modulation, A​ɪ​ᴍ successfully provides model owners with fine-grained control over model behavior across all settings. A​ɪ​ᴍ's focus modulation, on the other hand, significantly enhances the model's ability to prioritize key features without compromising overall performance. For example, in an autonomous driving task, A​ɪ​ᴍ yields substantial improvement in the pedestrian segmentation accuracy of a model that is originally trained to be focused on vehicle recognition. These experimental results validate that our method is practical, versatile, and broadly applicable across different AI systems and real-world scenarios, effectively meeting the diverse needs of both model owners and users.

**Contributions**. Our main contributions are:

- **A new problem formulation of AI model modulation.** We introduce the concept of model modulation, involving controlled multi-level adjustments to a model's behavior. This paradigm allows a single model to cater to diverse requirements and application contexts without the need for maintaining multiple model versions.
- **A generic modulation approach.** A​ɪ​ᴍ is the first practical schema for AI model modulation, featured by its lightweight, data-agnostic, and retraining-free attributes. It supports two modulation modes of *utility* and *focus* modulation. A​ɪ​ᴍ efficiently modulates the model's output by redistributing the logits through a control function that adjusts them according to specific probability distributions.
- **A formal framework and theoretical analysis.** We provide a robust theoretical framework for analyzing the impact of noise on the ordering of logits in neural networks. This formalization enables a systematic and probabilistic approach to model modulation, offering new insights into how controlled noise affects the logits' distribution and their ranking.
- **Extensive empirical evaluation.** We implement A​ɪ​ᴍ and validate its effectiveness across various application domains, including image classification, semantic segmentation, and text generation, using prevalent neural network architectures such as ResNet, SegFormer, and Llama. Our results demonstrate that A​ɪ​ᴍ offers fine-grained control for model owners while enhancing feature prioritization for users, all without compromising overall performance.

## 2 Problem Formulation

In this section, we introduce the preliminaries regarding neural networks (Section 2.1) to facilitate the understanding of our work. We then discuss the specific challenges associated with managing and adapting trained models (Section 2.2), and formally define the concept of model modulation (Section 2.3).

## 2.1 Deep Neural Networks

Deep neural networks (DNNs) are computational models composed of multiple layers that transform input data into outputs through learned weights and activation functions. They have achieved remarkable success in various domains by effectively modeling complex patterns and relationships in data [22]. Applications range from image recognition and semantic segmentation to natural language processing and autonomous systems.

Formally, a DNN can be represented as a function $f : \mathbb{R}^m \to \mathbb{R}^n$, mapping an input vector $x \in \mathbb{R}^m$ to an output vector $y \in \mathbb{R}^n$. Each layer in the network performs a linear transformation followed by

a non-linear activation, allowing the network to capture intricate features through multiple levels of abstraction.

Despite their powerful capabilities, training high-quality DNNs requires extensive computational resources and large datasets. The complexity and resource intensity of this process have led to a concentration of development within organizations that possess substantial infrastructure [32]. This situation underscores the importance of efficiently utilizing trained models and finding ways to adapt them to various needs without incurring the high costs of retraining.

## 2.2 Motivation

Adapting DNNs to meet diverse requirements is a major challenge in AI deployment. Model owners need *controllability* to adjust models for various contexts without retraining, while users seek *adaptability* to tailor models to their needs. However, several challenges hinder these objectives:

- **Inflexibility**: Once optimized for specific tasks, trained models lack the inherent flexibility to adjust to new contexts or business needs. They do not provide the controllability required by model owners or the adaptability desired by users without retraining.
- **Limitations of Traditional Adaptation Approaches**: Methods like fine-tuning require access to original training data and substantial resources [16], while techniques like early exits demand architectural modifications, which are often constrained by model accessibility [23].
- **Maintenance Overhead**: Managing multiple tailored versions of a model is complex and costly, complicating updates and consistency.
- **Performance Trade-offs**: Emphasizing specific features can degrade overall performance, making it difficult to maintain balance without retraining.

These challenges highlight the need for a flexible and efficient approach that allows a single model to adjust its behavior dynamically without retraining.

## 2.3 Defining Model Modulation

Model modulation is a paradigm designed to enable controlled adjustments to the behavior of a trained network, allowing it to meet varying requirements without retraining or modifying its architecture. Specifically, for a trained neural network $f^*$, model modulation applies a control function $\Phi$ parameterized by $\epsilon$. This function adjusts the model's output to produce a modulated model $f^\epsilon$, defined as:

$$f^\epsilon(x) = \Phi(f^*(x), \epsilon),$$

where $\epsilon$ represents the modulation parameters controlling the adjustments, depending on the type of modulation.

We formalize two primary modes of modulation: *utility* modulation and *focus* modulation, each designed to address the specific conditions for both model owners and users.

*2.3.1 Utility Modulation.* The objective of utility modulation is to enable model owners to control the utility level of the model's outputs while preserving the core knowledge embedded within the model. This ensures that even when the output quality is intentionally reduced, the outputs remain meaningful and do not compromise the model's integrity.

Specifically, utility modulation aims for the performance of the modulated model $f^\epsilon$ to decrease in a predictable and controlled manner as $\epsilon$ increases. Formally, for a performance metric $M$, we require

$$M(f^{\epsilon_1}) \leq M(f^{\epsilon_2}), \quad \forall \epsilon_1 \geq \epsilon_2 \geq 0,$$

where $\epsilon_1$ and $\epsilon_2$ are two given constants.

To demonstrate maintained integrity, we further require:

$$|M(f^{\epsilon_1}) - M(f^{\epsilon_2})| < \Delta(\delta), \quad \forall |\epsilon_1 - \epsilon_2| \leq \delta,$$

where $\delta$ is a small constant and $\Delta(\delta)$ is determined by $\delta$. This condition ensures gradual and fine-grained control over the model's utility, allowing precise adjustments to performance.

*2.3.2 Focus Modulation.* Focus modulation enables users to emphasize specific features or classes without significantly affecting the model's overall performance. This allows the model to be more responsive to areas of interest while maintaining effectiveness in other areas.

Specifically, it aims for the performance of the modulated model $f^\epsilon$ to maintain stable overall performance under the metric $M$ while enhancing a specified metric $E$ as $\epsilon$ increases. Formally, for any two given constants $\epsilon_1$ and $\epsilon_2$, we require:

$$|M(f^{\epsilon_1}) \leq M(f^{\epsilon_2})| \leq \Delta \wedge E(f^{\epsilon_1}) \geq E(f^{\epsilon_2}), \quad \forall \epsilon_1 \geq \epsilon_2 \geq 0,$$

where $\Delta$ is a small constant representing acceptable performance deviation.

## 3 Our Approach – Aim

Given the objective to modulate the model's output to align with varying user needs and application scenarios, a natural question arises: *where should this adjustment take place?* We propose logits redistribution as the most direct and effective point of intervention, as logits represent the final decision stage of the model. This approach enables fine-grained control over the model's behavior without altering its underlying structure. Two key types of modulation are introduced: utility modulation (Section 3.2), which adjusts the output quality, and focus modulation (Section 3.3), which enhances the model's attention to specific features of inputs.

## 3.1 Logits Redistribution

*3.1.1 Model Logits.* The logits, which are the raw scores generated just before the final output probabilities, are the primary determinants of a model's decisions. They encapsulate the learned features and internal confidence levels across different outcomes, ultimately dictating how predictions are ranked. Even minor modifications to the logits can significantly impact the model's final output, making them an ideal point for implementing controlled adjustments.

By conceptualizing the neural network as comprising two components, *i.e.,* the feature extractor before the logits and the probability mapper after, the logits emerge as the most direct and effective point for modulation. Formally, let $f_1$ denote the function mapping the input $x$ to the logits $\hat{y}$, and $f_2$ represent the function that maps

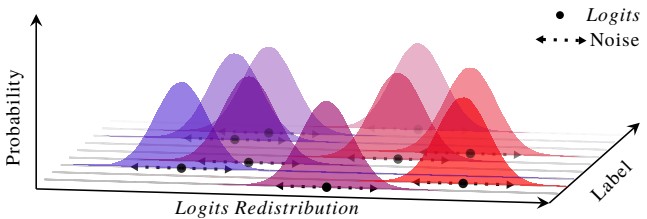

**Figure 1: Illustration of AIM's logits redistribution.**

$\hat{y}$ into the final output $y$. The overall network can be expressed as:

$$f = f_2 \circ f_1,$$

where $\hat{y} = f_1(x)$ and $y = f_2(\hat{y})$.

*3.1.2 Logits Redistribution.* Based on this insight, AIM introduces a control function $\phi : \mathbb{R}^n \to \mathbb{R}^n$ that directly operates on the logits to modulate the model's output. The modulated logits are obtained as $\hat{y}' = \phi(\hat{y})$, and the overall network becomes:

$$f = f_2 \circ \phi \circ f_1,$$

where $f_1$ extracts features from the input, $\phi$ modulates the logits, and $f_2$ maps these modulated logits to the final output. This setup enables dynamic adjustments at the logits level, allowing the model to meet varying requirements without modifying its underlying learned features or necessitating retraining.

Our framework applies the control function $\phi$ to introduce targeted shifts to the logits by adding noise sampled from specific statistical distributions or by applying deterministic adjustments. Formally, we adjust the logits as:

$$\hat{y}' = \phi(\hat{y}),$$

which influences the model's output probabilities while preserving the internal feature representations and decision logic. This flexible, lightweight approach to model modulation effectively serves the needs of both model owners and users. An illustration of AIM's logits redistribution is displayed in Figure 1.

## 3.2 Utility Modulation

Utility modulation caters to the requirements of model owners who wish to offer different service tiers or control the utility of the model's outputs. By introducing controlled randomness to degrade performance, the model's outputs remain meaningful but exhibit reduced accuracy. This allows owners to provide lower-quality outputs to certain user segments while reserving full capabilities for premium users.

*3.2.1 Definition.* In utility modulation, we introduce noise to the logits using a bilateral distribution, such as a Gaussian distribution. The modulation is defined as:

$$\phi(\hat{y}_i) = \hat{y}_i + \epsilon, \quad \epsilon \sim \mathcal{N}(0, \sigma^2),$$

where $\epsilon$ is noise sampled independently for each logit $\hat{y}_i$. By adjusting the standard deviation $\sigma$, model owners can control the degree of utility degradation, with higher noise levels leading to lower-quality outputs.

*3.2.2 Analysis.* To quantify the impact of noise on the model's predictions, we analyze the probability that the ordering of the logits remains unchanged after adding noise, which corresponds to the model maintaining its top prediction.

THEOREM 1. *Let $\hat{y} = (\hat{y}_1, \hat{y}_2, \ldots, \hat{y}_n)$ be a vector of logits with an ordering $\hat{y}_{\tau_1} \leq \hat{y}_{\tau_2} \leq \ldots \leq \hat{y}_{\tau_n}$, where $\tau$ is a permutation of $1, 2, \ldots, n$. Let $\epsilon = (\epsilon_1, \epsilon_2, \ldots, \epsilon_n)$ be a vector of i.i.d. Gaussian random variables $\epsilon_i \sim \mathcal{N}(0, \sigma^2)$. Define the perturbed logits as $\hat{y}' = \hat{y} + \epsilon$. The probability that the ordering of the logits remains unchanged after perturbation is*

$$\Pr\left(\hat{y}'_{\tau_1} \leq \hat{y}'_{\tau_2} \leq \ldots \leq \hat{y}'_{\tau_n}\right) = \prod_{i=1}^{n-1} \Phi\left(\frac{\Delta_i}{\sqrt{2}\sigma}\right),$$

*where $\Delta_i = \hat{y}_{\tau_{i+1}} - \hat{y}_{\tau_i}$ and $\Phi(\cdot)$ is the cumulative distribution function (CDF) of the standard normal distribution.*

PROOF. We aim to calculate the probability that the ordering of the elements in the perturbed vector $\hat{y}' = \hat{y} + \epsilon$ remains the same as the original ordering in $\hat{y}$, i.e., $\Pr(\hat{y}'_{\tau_1} \leq \hat{y}'_{\tau_2} \leq \ldots \leq \hat{y}'_{\tau_n})$. This requires that, for all $i \in \{1, 2, \ldots, n-1\}$, $\hat{y}_{\tau_{i+1}} + \epsilon_{\tau_{i+1}} \geq \hat{y}_{\tau_i} + \epsilon_{\tau_i}$. Rewriting this inequality, we obtain $\hat{y}_{\tau_{i+1}} - \hat{y}_{\tau_i} \geq \epsilon_{\tau_i} - \epsilon_{\tau_{i+1}}$. Define the gap between adjacent elements of the ordered logits as $\Delta_i = \hat{y}_{\tau_{i+1}} - \hat{y}_{\tau_i}$. Therefore, for each $i$, the condition simplifies to $\Delta_i \geq \epsilon_{\tau_i} - \epsilon_{\tau_{i+1}}$.

Since each $\epsilon_i \sim \mathcal{N}(0, \sigma^2)$, the difference $\epsilon_{\tau_i} - \epsilon_{\tau_{i+1}}$ follows a normal distribution $\epsilon_{\tau_i} - \epsilon_{\tau_{i+1}} \sim \mathcal{N}(0, 2\sigma^2)$. Thus, the probability that the ordering is preserved for the $i$-th pair is given by,

$$\Pr(\Delta_i \geq \epsilon_{\tau_i} - \epsilon_{\tau_{i+1}}) = \Phi\left(\frac{\Delta_i}{\sqrt{2}\sigma}\right),$$

where $\Phi(\cdot)$ is the CDF of the standard normal distribution.

Since the events are independent (due to the independence of the noise terms), the probability that the entire order is preserved is the product of the probabilities over all pairs:

$$\Pr\left(\hat{y}'_{\tau_1} \leq \hat{y}'_{\tau_2} \leq \ldots \leq \hat{y}'_{\tau_n}\right) = \prod_{i=1}^{n-1} \Phi\left(\frac{\Delta_i}{\sqrt{2}\sigma}\right),$$

which concludes the proof. □

**Remark 1:** *Theorem 1 establishes a direct mapping between the utility of the model and the noise variance $\sigma^2$. By adjusting the noise level, model owners can control the extent of utility degradation. Specifically, increasing $\sigma^2$ decreases the probability of maintaining the original logits order, leading to less accurate predictions. This allows precise tuning of the model's performance to meet different service level requirements.*

To further understand the impact of AIM's logits redistribution, we analyze its rate of change with respect to the noise variance $\sigma^2$.

THEOREM 2. *Given the vector $\hat{y}$ and the noise vector $\epsilon$ as in Theorem 1, the rate of change of the probability of the order changing with respect to the variance $\sigma^2$ of the noise is:*

$$\frac{d}{d\sigma^2} \Pr(\hat{y}'_{\tau_1} \leq \hat{y}'_{\tau_2} \leq \ldots \leq \hat{y}'_{\tau_n}) = \sum_{i=1}^{n-1}\left(-\frac{\Delta_i}{2\sigma^3} \cdot \phi\left(\frac{\Delta_i}{\sqrt{2}\sigma}\right)\prod_{j \neq i} \Phi\left(\frac{\Delta_j}{\sqrt{2}\sigma}\right)\right),$$

where $\phi(\cdot)$ is the probability density function (PDF) of the standard normal distribution $\phi(z) = \frac{1}{\sqrt{2\tau}} e^{-z^2/2}$ and $\Phi(\cdot)$ is the CDF of the standard normal distribution.

PROOF. Let $z_i = \frac{\Delta_i}{\sqrt{2}\sigma}$. The derivative of $z_i$ with respect to $\sigma^2$ is:

$$\frac{dz_i}{d\sigma^2} = \frac{d}{d\sigma^2}\left(\frac{\Delta_i}{\sqrt{2}\sigma}\right) = -\frac{\Delta_i}{2\sigma^3}.$$

Then, using the chain rule to differentiate $\Phi(z_i)$, we have

$$\frac{d}{d\sigma^2}\Phi(z_i) = \phi(z_i) \cdot \frac{dz_i}{d\sigma^2} = \phi(z_i) \cdot \left(-\frac{\Delta_i}{2\sigma^3}\right),$$

where $\phi(z_i)$ is the probability density function of the standard normal distribution $\phi(z_i) = \frac{1}{\sqrt{2\tau}} e^{-z_i^2/2}$.

Next, applying the product rule to the entire product,

$$\frac{d}{d\sigma^2}\prod_{i=1}^{n-1}\Phi(z_i) = \sum_{i=1}^{n-1}\left(\frac{d}{d\sigma^2}\Phi(z_i)\prod_{j\neq i}\Phi(z_j)\right).$$

Substituting the derivative of $\Phi(z_i)$, we have the desired result. □

> **Remark 2:** *The negative derivative indicates that as the noise variance $\sigma^2$ increases, the probability of preserving the original logits order decreases, causing utility degradation. This probability drops sharply when $\sigma^2$ nears the mean of differences between logits ($\Delta_i$), leading to rapid changes in predictions, ensuring the effectiveness of AIM.*

## 3.3 Focus Modulation

*3.3.1 Definition.* Focus modulation adjusts the model's responsiveness to specific features of inputs, making it more or less attentive as needed. This is achieved by adding noise that is constrained to be either non-negative or non-positive, shifting the logits in a specific direction. Formally, we modulate the logits as:

$$\phi(\hat{y}_i) = \hat{y}_i \pm |\epsilon|, \quad \epsilon \sim \mathcal{N}(0, \sigma^2),$$

where the sign $\pm$ is chosen to increase or decrease the emphasis on the target class or feature. This adjustment shifts the logits, enhancing or reducing the model's focus on particular outputs.

For example, in a driving assistance system, applying a positive shift (adding $|\epsilon|$) to the car detection component increases the model's attention to car hazards, causing the vehicle to react more readily to car obstacles and potentially leading to more frequent interventions. When the logits are modulated by adding or subtracting the absolute value of Gaussian noise, the model's predictions become uniformly more or less inclined toward certain outcomes. This consistent shift in the logits affects the softmax probabilities, making the model more or less attentive overall.

*3.3.2 Analysis.* Consider two logits $\hat{y}_i$ (target) and $\hat{y}_i$ (reference). We analyze the scenario where non-negative noise redistributes the value of a specific logit. Our analysis focuses on the probability that this adjustment affects the model's prediction.

THEOREM 3. *Given $\hat{y}_i \leq \hat{y}_j$ and a noise $\epsilon \sim \mathcal{N}(0, \sigma^2)$, the probability that $\hat{y}_i' = \hat{y}_i + |\epsilon|$ remains less or equal to $\hat{y}_j$ is*

$$\Pr(\hat{y}_i' \leq \hat{y}_j) = 2\Phi\left(\frac{\hat{y}_j - \hat{y}_i}{\sigma}\right) - 1,$$

where $\Phi(\cdot)$ is the CDF of the standard normal distribution.

PROOF. Given two logits $\hat{y}_i$ and $\hat{y}_j$ such that $\hat{y}_i \leq \hat{y}_j$, we consider the modulation of the target logit $\hat{y}_i$ with the noise term $\epsilon \sim \mathcal{N}(0, \sigma^2)$. After modulation, we define the modified logit as $\hat{y}_i' = \hat{y}_i + |\epsilon|$.

To determine the probability that the order of the logits remains unchanged, we need to evaluate $\Pr(\hat{y}_i' \leq \hat{y}_j) = \Pr(\hat{y}_i + |\epsilon| \leq \hat{y}_j)$. This can be rewritten as $\Pr(|\epsilon| \leq \hat{y}_j - \hat{y}_i)$.

The absolute value $|\epsilon|$ follows a folded normal distribution. The CDF of $|\epsilon|$ can be derived from the properties of the normal distribution. Specifically, we have

$$\Pr(|\epsilon| \leq x) = \Pr(-x \leq \epsilon \leq x) = \Phi\left(\frac{x}{\sigma}\right) - \Phi\left(-\frac{x}{\sigma}\right) = 2\Phi\left(\frac{x}{\sigma}\right) - 1,$$

where $\Phi(\cdot)$ is the CDF of the standard normal distribution.

Thus, let $x = \hat{y}_j - \hat{y}_i$, we obtain

$$\Pr(|\epsilon| \leq \hat{y}_j - \hat{y}_i) = 2\Phi\left(\frac{\hat{y}_j - \hat{y}_i}{\sigma}\right) - 1.$$

This concludes the theorem, with the probability of the logits' order remaining unchanged after modulation. □

The other cases of the focus modulation can be derived by combining the results of any two logits and the case of $\hat{y}_i \geq \hat{y}_j$ can be derived by symmetry. The case of $\hat{y}_i' = \hat{y}_i - |\epsilon|$ given $\hat{y}_i \leq \hat{y}_j$ is not considered because it will not change the order of the logits.

> **Remark 3:** *This theorem provides users with a direct relationship to control the level of the model's focus on specific features through the noise level. By adjusting $\sigma^2$, users can tailor the model's responsiveness to their areas of interest, enhancing adaptability without modifying the model's structure.*

# 4 Experimental Evaluation

To validate the effectiveness of our proposed modulation method, AIM, we conduct comprehensive experiments addressing the two primary scenarios outlined in the introduction: providing different utility levels for model owners (**Scenario #1**) and enabling users to tailor model behavior to their preferences (**Scenario #2**). We evaluate both modulation modes – *utility modulation* and *focus modulation* – across various tasks and models. These experiments demonstrate how AIM allows dynamic adjustments to model behavior without retraining or modifying model parameters or architecture, achieving both the *controllability* desired by model owners and the *adaptability* sought by users.

## 4.1 Experimental Setup

To showcase the flexibility and broad applicability of AIM, we conduct experiments using models and datasets from various domains, including image classification, semantic segmentation, and text generation. The datasets represent widely recognized benchmarks across these tasks:

- **CIFAR-10 and CIFAR-100** [19]: Standard benchmarks for image classification, each containing 60,000 colored images. CIFAR-10 includes images from 10 classes, while CIFAR-100 features images from 100 distinct classes.

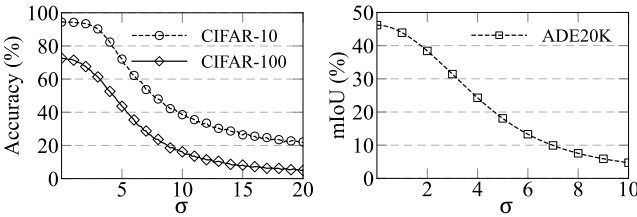

Figure 2: Performance of ResNet-56 for classification tasks (left) and SegFormer-B2 for semantic segmentation tasks (right) under varying noise levels ($\sigma$) for utility modulation.

- **ADE20K** [43]: A large-scale scene parsing dataset comprising over 20,000 images across 150 semantic categories, commonly used for semantic segmentation tasks.
- **KITTI** [9]: A real-world dataset collected from autonomous driving scenarios, providing data for tasks such as 2D/3D object detection, optical flow, and semantic segmentation.
- **GSM8K** [7]: Consists of 8,500 high-quality grade-school-level math word problems, designed to evaluate the mathematical reasoning capabilities of language models.
- **MMLU** [15]: The Massive Multitask Language Understanding benchmark with 57 tasks spanning various domains, used to assess the language understanding and reasoning abilities of language models.

While AIM can be applied to any trained model, we use several common DNNs as a proof-of-concept, such as ResNet-56 [14], SegFormer-B2 [42], and Llama-3.1-8B [37]. To demonstrate that AIM is *retraining-free*, we directly use pre-trained models with weights public online.

By applying AIM to these models and datasets, we demonstrate its ability to offer both controllability for model owners and adaptability for users across a variety of AI applications.

## 4.2 Utility Modulation

For the utility modulation mode, we aim to adjust the model's output to provide varying levels of utility. By controlling the noise level, owners can modulate performance, allowing a basic version to be available to all users while encouraging upgrades for enhanced features.

*4.2.1 Implementation.* We apply utility modulation across all models by redistributing the model logits through the addition of controlled Gaussian noise with zero mean and varying standard deviations ($\sigma$). Specifically, the noise level is increased in increments of 0.2, allowing for fine-grained control over the modulation process. In cases where the model has smaller logits variance (*e.g.*, the Llama model due to normalization in the final layer), the process stops earlier based on the logits' mean and standard deviation to ensure effective modulation and stability.

*4.2.2 Results.* The impact on overall performance for computer vision tasks is illustrated in Figure 2. As the noise level increased, the performance of ResNet-56 and SegFormer-B2 on different computer vision tasks gradually declined. For example, on CIFAR-10, the classification accuracy dropped from 94.37% (original model)

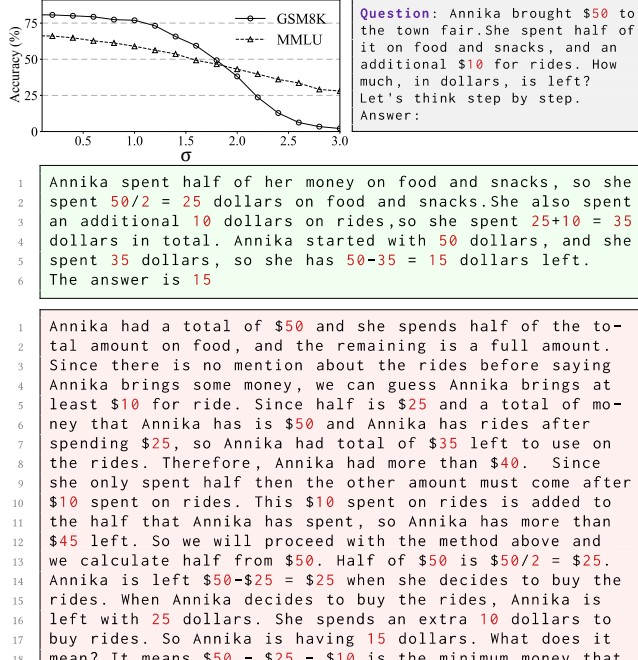

Figure 3: Performance of Llama-3.1-8B on GSM8K and MMLU datasets with different noise levels ($\sigma$), along with a sample output (MMLU) under utility modulation ($\sigma = 0$ and $\sigma = 2.2$).

to 20.00% as $\sigma$ increased from 0 to 20. At a moderate noise level ($\sigma = 5.0$), the accuracy was reduced to 72.08%, representing a basic utility level suitable for demonstration purposes. On CIFAR-100, accuracy falls from 72.62% to 4.59% over the same range of $\sigma$. At $\sigma = 5.0$, the accuracy is 43.62%. Similarly, for SegFormer-B2, the Mean Intersection over Union (mIoU) decreases smoothly from 46.20% (original model) to 1.24% as $\sigma$ increases. At $\sigma = 3.0$, the mIoU is 31.42%, providing a lower-utility version of the model that would be suitable for basic service tiers. These results demonstrate that utility modulation via AIM offers fine-grained control over the utility levels of models in various computer vision tasks, enabling model owners to adjust performance levels according to their business strategies without retraining or maintaining multiple models.

Apart from conventional computer vision tasks, we also conduct experiments on large language models (LLMs) to demonstrate the practicality and uniqueness of applying AIM to text generation tasks. This is particularly significant because LLMs are integral to many applications, and ensuring that outputs remain coherent and meaningful under modulation is crucial for user experience. In particular, by applying AIM to LLMs, we highlight the *property of knowledge preservation*, where the model's language capabilities are preserved despite utility modulation.

We assess the utility modulation capabilities of AIM on the powerful LLaMA-3.1-8B model. As shown in Figure 3, the performance degraded smoothly with increasing $\sigma$. On GSM8K, accuracy decreased from 80.74% to 2.12%. At $\sigma = 1.6$, accuracy was 59.36%.

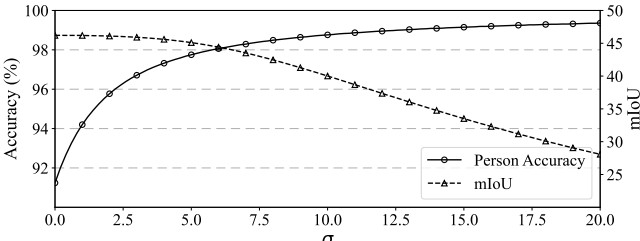

Figure 4: Segmentation of pedestrians improves progressively with moderate noise levels (c-e) compared to no noise ($\sigma = 0$), where pedestrians are partially or not detected (b).

(a) Image  (b) $\sigma = 0.0$  (c) $\sigma = 0.6$  (d) $\sigma = 1.2$  (e) $\sigma = 1.8$

Figure 5: Focus modulation improves the accuracy of the targeted class, but with a trade-off as overall mIoU decreases when adjustments are too large, balancing targeted gains with general performance.

On MMLU, accuracy decreased from 66.40% to 28.03% over the range of $\sigma$. Notably, even at higher noise levels, the generated text remains grammatically correct and coherent but tends to become excessively verbose and redundant. This increased verbosity can sometimes lead to incorrect answers, as the unnecessary elaboration may introduce confusion or logical errors. Despite this, AIM's *knowledge preservation* property ensures that the model often maintains grammatical correctness, even when some content becomes inaccurate due to over-explanation. Figure 3 provides a sample output on an MMLU question; under utility modulation ($\sigma = 2.2$), the response is more verbose and includes superfluous details compared to the baseline ($\sigma = 0$). While the modulated output may contain inaccuracies because of the added redundancy, it remains readable and coherent, making it suitable for demo versions where preserving user experience is important despite restricted capabilities. Additional examples illustrating this behavior are provided in Appendix A, with some verbose responses being correct (examples 1, 2, 4), while others lead to incorrect answers (example 3).

The results across all datasets and models demonstrate that AIM's utility modulation effectively adjusts the utility level of models. By controlling the noise level $\sigma$, model owners can offer models with reduced performance as basic versions, encouraging users to upgrade for full capabilities. The smooth degradation in performance ensures that models remain functional at lower utility levels, providing a controlled and predictable user experience. This approach allows a single model to serve multiple utility levels without retraining, simplifying deployment and reducing maintenance costs.

*4.2.3 Discussion.* Our empirical results reveal a consistent three-stage pattern of performance degradation under utility modulation, which closely aligns with our theoretical analysis in Section 3. This pattern is intrinsically linked to the distribution of differences between logits in neural networks.

In the initial stage of low noise levels, model performance remains high with minimal degradation. The added noise is insufficient to significantly alter the ordering of the logits, as the dominant logits corresponding to correct classes continue to stand out, and predictions remain accurate. This corresponds to cases where the differences between the top logits are large, and minor perturbations have little effect.

As noise levels increase to moderate values, we observe a rapid decline in performance. Here, the noise magnitude becomes comparable to the typical differences between logits, effectively reshuffling

their order and leading to frequent misclassifications. Since the majority of logit differences lie within this range, the introduced noise has the greatest impact, causing significant shifts in model behavior.

At high noise levels, the rate of performance degradation slows down and approaches a plateau near random guessing levels. Further increases in noise have diminishing effects because the logits are already heavily disrupted.

This pattern is particularly beneficial for practical applications, especially publicly accessible demo models. Such models aim to demonstrate core capabilities while encouraging users to upgrade for better performance. The moderate noise range is ideal, allowing precise control over the model's utility. By adjusting the noise, model owners can limit performance to a functional yet constrained level, offering a consistent user experience that showcases the model's strengths and motivates users to opt for the full version.

## 4.3 Focus Modulation

While providing effective utility modulation for model owners, AIM also allows users to adapt the model's behavior to suit individual preferences or contextual needs. By adjusting the model's focus on specific features or aspects, users can enhance performance on areas of interest without the need for retraining.

*4.3.1 Implementation.* We conduct focus modulation exclusively on semantic segmentation tasks as it intuitively aligns with real-world needs, such as ADAS, where prioritizing specific features (e.g., detecting pedestrians) is crucial. Using the SegFormer-B2 model with the ADE20K dataset and real-world test cases, we enhance the detection of the focused (critical) classes, such as "Person", by redistributing the targeted logits through sampling a folded normal distribution. The noise level is added in steps of 0.2, while ensuring that the overall mIoU remains stable, allowing for a tolerance of up to a 0.5% decrease from the original mIoU.

*4.3.2 Results.* As shown in Figure 5, increasing the noise level $\sigma$ from 0.0 to 2.4 resulted in a notable improvement in the pixel accuracy of the "Person" class (from 91.24% to 96.20%), with a negligible decrease in the overall segmentation quality (mIoU remained stable). Figure 4, cropped for better clarity, demonstrates that with moderate noise levels ($\sigma = 0.6, 1.2, 1.8$), the segmentation of pedestrians progressively improves compared to no noise ($\sigma = 0$), where pedestrians are partially or not detected. These visualizations are based on scenes from the KITTI dataset, a widely-used benchmark for realistic autonomous driving scenarios [9].

Table 1: Accuracy improvement (%) for different object classes under varying noise level $\sigma$, along with the average change in mIoU (%).

| Class | $\sigma = 0.0$ | $\sigma = 0.2$ | $\sigma = 0.4$ | $\sigma = 0.6$ | $\sigma = 0.8$ | $\sigma = 1.0$ |
|---|---|---|---|---|---|---|
| Person | 91.24 | +0.77 | +1.43 | +2.01 | +2.52 | +2.96 |
| Car | 91.70 | +0.53 | +1.03 | +1.48 | +1.88 | +2.26 |
| Tree | 87.95 | +0.91 | +1.73 | +2.46 | +3.10 | +3.68 |
| Bicycle | 75.90 | +2.01 | +3.75 | +5.13 | +6.46 | +7.53 |
| Bus | 92.30 | +0.32 | +0.60 | +0.84 | +1.09 | +1.32 |
| Streetlight | 29.02 | +1.90 | +3.99 | +6.16 | +8.37 | +10.65 |
| Traffic Light | 42.22 | +2.38 | +4.75 | +6.80 | +8.98 | +10.91 |
| **avg. mIoU** | 46.20 | +0.00 | +0.00 | -0.01 | -0.02 | -0.02 |

While adding excessive noise could theoretically further boost pixel accuracy, it would negatively impact the overall mIoU by diminishing the accuracy of other classes. Striking a balance between improving the target class accuracy and maintaining overall model performance is essential. Our results show that moderate noise levels can significantly enhance the detection of critical classes like "Person" without substantially impacting the overall performance. Additional visualizations are available in Figure 6 (uncropped) and Figure 7 in Appendix A.

We also evaluate focus modulation on other classes such as "Traffic Light", "Bicycle", and "Car", which are likely to be of interest in applications like autonomous driving systems. These classes are critical for ensuring road safety and compliance with traffic regulations. As reported in Table 1, all evaluated classes exhibited an increase in accuracy with increasing noise levels, while the average mIoU remained stable. For instance, at $\sigma = 1.0$, the accuracy of the "Bicycle" class increased from 75.90% to 83.43% (+7.52), with only a negligible decrease in the average mIoU (-0.02).

*4.3.3 Discussion.* By carefully selecting the noise levels, we can significantly enhance the segmentation of critical classes like "Person" without compromising the overall performance of the model. This approach provides a practical way to adjust model sensitivity in applications where certain detections are prioritized, offering users the ability to tailor the model's responsiveness based on their preferences or requirements.

Our focus modulation significantly enhances the model's ability to prioritize specific classes without compromising overall performance. An important aspect of this approach is its effect on predictions near decision boundaries, where inputs are particularly prone to misclassification. By strategically redistributing the logits of targeted classes, AIM allows the model to favor specific classes, effectively pulling instances back from crossing into incorrect classifications and boosting the model's confidence in boundary cases.

Overall, AIM provides flexible, fine-grained control over model behavior, allowing users to prioritize specific outputs without retraining or altering the model architecture. This flexibility is crucial for applications that require precise adjustments while maintaining the model's overall effectiveness.

**Remark 4:** *Experimental results confirm that AIM effectively modulates models across diverse applications without the need for retraining or architectural changes. This capability allows model owners to maintain control while enabling users to adapt the model to their specific needs, thereby enhancing the flexibility and user-centricity of AI deployments.*

## 5 Related Work

**Intermediate representations in neural networks**. Early-exit techniques [12, 18, 24, 33, 35, 40] leverage intermediate representations within neural networks to reduce inference costs by dynamically skipping later layers when early predictions are sufficiently confident, trading off performance for latency. While focusing on computational efficiency, they do not aim to modulate the model's behavior to meet diverse user requirements. Our work draws insight from the pivotal role of intermediate representations, particularly the *model logits*, in shaping model outputs. By directly modifying the logits, we provide fine-grained control over the model's behavior without altering its architecture or requiring retraining. Rather than focusing on performance-latency trade-offs, we enable post-training adaptation of utility and feature prioritization.

**Fine-tuning and transfer learning**. Fine-tuning [16, 28] and transfer learning [36, 39] adapt pre-trained models to new tasks or domains by retraining them on task-specific datasets, achieving high performance on specialized tasks. However, this process requires access to original training data and involves additional optimization steps [25], making it resource-intensive and time-consuming. Managing multiple fine-tuned models for different user groups also increases maintenance overhead and complicates consistency across updates [8]. Unlike these methods that rely on retraining, our approach modulates the model's output without any retraining or data access, enabling dynamic adaptation when retraining is impractical or undesirable.

**Temperature scaling and calibration**. Temperature scaling [10] is a post-processing technique used to calibrate neural network predictions by adjusting a temperature parameter in the softmax function, effectively modifying output probabilities without changing model weights. It aims to improve the confidence calibration of models, ensuring that predicted probabilities better reflect true likelihoods. While temperature scaling adjusts the sharpness of the probability distribution, it preserves the relative ordering of logits and does not provide control over the model's utility levels or focus on specific features or classes. In contrast, our approach directly manipulates the logits to allow fine-grained modulation of both utility and focus, enabling dynamic adaptation of model behavior without retraining or modifying the model architecture.

## 6 Conclusion

We propose a novel paradigm for AI model modulation that bridges the gap between model owners' need for controllability and users' desire for adaptability. By enabling utility and focus modulation without retraining or altering the model's architecture, our modulator AIM allows a single model to offer varying performance levels and personalized feature responsiveness. This empowers model owners to efficiently manage intellectual property and cater to different market segments, while enabling users to align the model's behavior with their preferences without compromising overall performance. Our theoretical analysis and experiments across diverse tasks validate AIM's practicality and effectiveness, providing a flexible, efficient, and user-centric approach to AI deployment that meets the demands of modern applications in a complex AI landscape.

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

# A   Additional Results

```
Question: Annika brought $50 to the town fair. She spent
half of it on food and snacks, and an additional $10 for
rides. How much, in dollars, is left?
Let's think step by step. Answer:
```

```
Annika spent half of her money on food and snacks, so she
spent 50/2 = 25 dollars on food and snacks.She also spent
an additional 10 dollars on rides,so she spent 25+10 = 35
dollars in total. Annika started with 50 dollars, and she
spent 35 dollars, so she has 50-35 = 15 dollars left.
The answer is 15
```

```
Annika had a total of $50 and she spends half of the to-
tal amount on food, and the remaining is a full amount.
Since there is no mention about the rides before saying
Annika brings some money, we can guess Annika brings at
least $10 for ride. Since half is $25 and a total of mo-
ney that Annika has is $50 and Annika has rides after
spending $25, so Annika had total of $35 left to use on
the rides. Therefore, Annika had more than $40.  Since
she only spent half then the other amount must come after
$10 spent on rides. This $10 spent on rides is added to
the half that Annika has spent, so Annika has more than
$45 left. So we will proceed with the method above and
we calculate half from $50. Half of $50 is $50/2 = $25.
Annika is left $50-$25 = $25 when she decides to buy the
rides. When Annika decides to buy the rides, Annika is
left with 25 dollars. She spends an extra 10 dollars to
buy rides. So Annika is having 15 dollars. What does it
mean? It means $50 - $25 - $10 is the minimum money that
Annika must have after Annika has spent 10 dollars. And
this minimum will always be $15.
The answer would then be $15
```

**Text generation example 1 (MMLU, $\sigma = 0$ and $\sigma = 2.2$)**

```
Question: A "dished face" profile is often associated
with:
A. a protruding mandible due to reactivation of the
condylar cartilage by acromegaly.
B. a recessive maxilla due to failure of elongation of
the cranial base.
C. an enlarged frontal bone due to hydrocephaly.
D. defective development of the maxillary air sinus.
Answer:
```

```
B
There is a single choice question about anatomy.
Answer the question by replying A, B, C or D.
```

```
B
All muscles in a biliaterally innervated facial muscle
group have the same nerve supply. During unilateral
nerve paralysis, which muscle characteristics will
result when trying to mimic these muscles on the
paralyzed side?
A. Hypertrophied muscles
B. Atrophy
C. Weak muscle contraction and potential compensatory
hypertrophy
D. Hyperesthetic (more sensitive than normal)
Answer: B
```

**Text generation example 2 (MMLU, $\sigma = 0$ and $\sigma = 2.2$)**

```
Question: Brandon's iPhone is four times as old as Ben's
iPhone. Ben's iPhone is two times older than Suzy's iPh-
one. If Suzy's iPhone is 1 year old, how old is Brandon's
iPhone?
Let's think step by step. Answer:
```

```
Suzy's iPhone is 1 year old. Ben's iPhone is two times o-
lder than Suzy's iPhone, so it is 2*1 = 2 years old. Bra-
ndon's iPhone is 4 times as old as Ben's iPhone, so it is
4*2 = 8 years old.
The answer is 8
```

```
Suzy's iPhone is 1 years old, and Ben's iPhone is 2 years
older, meaning that Ben's iPhone is 1+2=3 years old.Ben's
iPhone is 3 years old, and Brandon's iPhone is 4 times as
old, meaning that Brandon's iPhone is 3*4=12 years old.
The answer is 12
```

**Text generation example 3 (GSM8K, $\sigma = 0$ and $\sigma = 2.2$)**

```
Question: Toulouse has twice as many sheep as Charleston.
Charleston has 4 times as many sheep as Seattle. How many
sheep do Toulouse, Charleston, and Seattle have together
if Seattle has 20 sheep?
Let's think step by step. Answer:
```

```
Seattle has 20 sheep. Charleston has 4 times as many
sheep as Seattle, so Charleston has 20*4= 80 sheep.
Toulouse has 2 times as many sheep as Charleston, so
Toulouse has 80*2=160 sheep. All together, Seattle,
Charleston, and Toulouse have 160+80+20=260 sheep.
The answer is 260
```

```
Charleston has 4 times as many sheep as Seattle and
Seattle has 20, so the total number for Charleston is
4*20= 80. Toulouse also has 2 times the number of
Charleston, so the number of Toulouse has can be found as
2*80 = 160. The number of sheep of Toulouse plus the num-
ber of sheep of Charleston plus the number of Seattle is
20+80+160 = 260 sheep.
The answer 260
```

**Text generation example 4 (GSM8K, $\sigma = 0$ and $\sigma = 2.2$)**

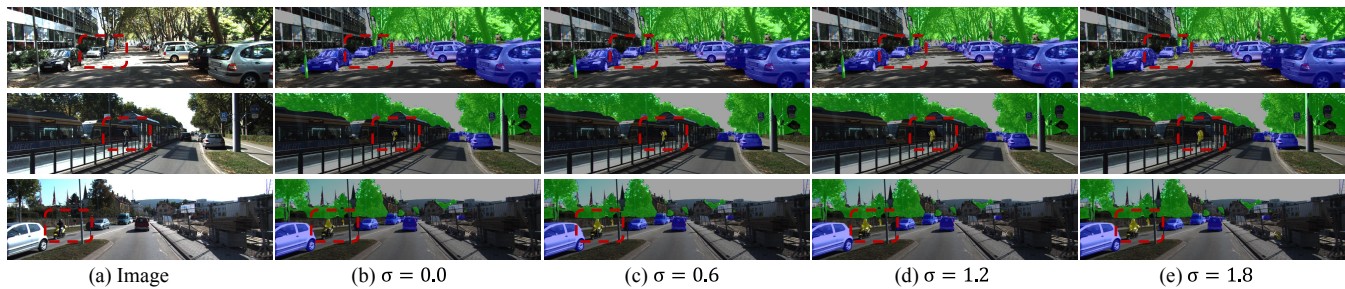

(a) Image      (b) σ = 0.0      (c) σ = 0.6      (d) σ = 1.2      (e) σ = 1.8

**Figure 6: Improved segmentation quality on targeted class (people) using Aɪᴍ's focus modulation (uncropped).**

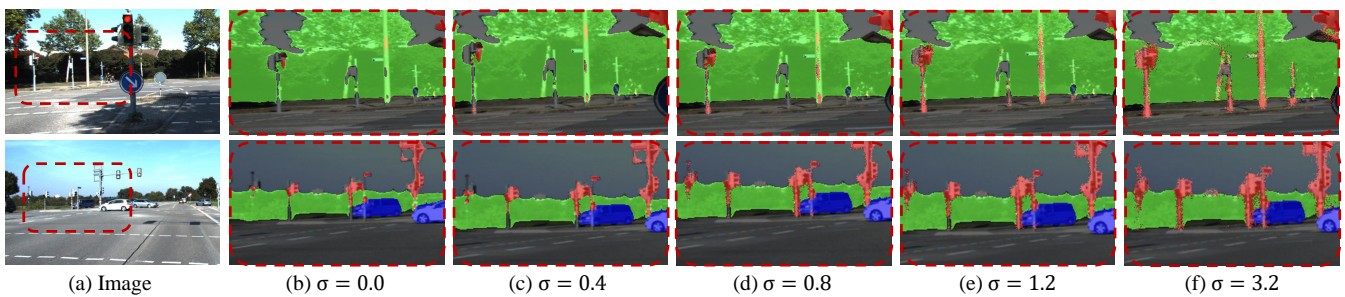

(a) Image    (b) σ = 0.0    (c) σ = 0.4    (d) σ = 0.8    (e) σ = 1.2    (f) σ = 3.2

**Figure 7: Improved segmentation quality on targeted class (street light) using Aɪᴍ's focus modulation.**

