# OpenReview forum: "AI Model Modulation with Logits Redistribution"
_ACM.org/TheWebConf/2025/Conference — WWW 2025 Poster_

### Official Review · Reviewer_ygzy · 2024-11-20

**Novelty:** 3
**Technical Quality:** 4

**Review:**

This paper presents a paradigm for AI model modulation called Aim. This approach enables a single neural network to adapt its behavior to meet diverse requirements without retraining.

## Pros:
- The algorithm is straightforward and easy to implement.
- The theoretical analysis of the proposed framework is comprehensive.
- The empirical results are extensive, including evaluations on both large language models (LLMs) and image segmentation tasks.

## Cons:
- Despite the thorough theoretical and empirical analysis, the core idea of adding noise to logits lacks novelty.
- The proposed method has limited connection to web.
- Although adding noise would be a simple approach, distillation would be another approach which automatically provides models with different performances while reducing the inference speed with smaller model size. The paper lacks empirical comparisons with distillation-based approaches.
- The connection between the analysis of the proposed method and the model modulation section (2.3) is limited and could be better elaborated.

**Questions:**

- How do you ensure that adding noise would satisfy the three requirements on line 299, 303, and 318?
- Are the $\Delta$s on line 305 same as line 318?
- Is there a difference between equations on line 368 vs. line 380?
- Figure 1 needs more context. Does it depict the distribution of the noise?

**Reviewer Confidence:**

3: The reviewer is confident but not certain that the evaluation is correct

**Scope:**

2: The connection to the Web is incidental, e.g., use of Web data or API

---

### Official Review · Reviewer_FFJK · 2024-12-01

**Novelty:** 5
**Technical Quality:** 5

**Review:**

Advantages:
1. This paper introduces the novel concept of model modulation in artificial intelligence. Addressing this problem is highly significant, as training large AI models is resource-intensive, and modulation offers a promising solution to optimize resource usage.
2. The paper investigates a set of generic approaches for modulating AI models. These approaches are lightweight, requiring only a small number of parameters to be tuned, making them efficient and practical.
3. The authors conducted extensive experiments across both vision and text-based tasks. The results demonstrated the effectiveness of the proposed logits redistribution technique, showcasing its potential across diverse modalities.

Weaknesses:
1. In the utility modulation setting, the paper presents an interesting observation: with controlled noise injection, model performance decreases smoothly while preserving core knowledge. While this is promising, a more systematic evaluation of output coherence would strengthen the argument. For instance, a manual evaluation using a few hundred examples, followed by reporting a coherence score, would provide more insights into how well the outputs remain intelligible.
2. The focus modulation technique shows strong performance on a personalized ADAS dataset. However, to demonstrate its broader applicability, additional experiments on other tasks (e.g., emotion classification or persona-based conversational models) would help validate its generalizability across visual and language domains.
3. It would be beneficial to include baseline approaches for evaluating focus modulation in the context of personalization. This comparison could highlight the relative advantages of the proposed method.

**Questions:**

1. How did you determine the noise value used in the utility modulation experiments?
2. In real-world scenarios involving new users, how would you approach selecting an appropriate noise value or function to provide outputs tailored to them? Specifically, is it feasible to achieve this without prior knowledge about the users?

**Reviewer Confidence:**

3: The reviewer is confident but not certain that the evaluation is correct

**Scope:**

3: The work is somewhat relevant to the Web and to the track, and is of narrow interest to a sub-community

---

### Official Review · Reviewer_PKFn · 2024-12-01

**Novelty:** 6
**Technical Quality:** 4

**Review:**

The paper introduces Aim, a technique for adjusting AI models post-training to meet specific user needs without retraining. It allows for utility and focus adjustments, controlling performance and attention to input features. Theoretical analysis and experiments across various tasks confirm Aim’s effectiveness and efficiency, offering a user-centric AI deployment strategy.

pros:
1. this paper provides a flexible way to tailor a single model’s behavior to various user requirements, enhancing its applicability across diverse scenarios.
2. The paper addresses a significant challenge in the realm of large-scale deep learning by conceptualizing big models as a service. It introduces a novel approach to adapt the performance and behavior of a single, unified model to cater to various user needs, thereby advancing the concept of model-as-a-service.
3. The methodology presented in the paper is both straightforward and potent, involving the modulation of a model’s logits. This simplicity not only makes the technique easy to understand and implement but also demonstrates remarkable effectiveness in customizing model outputs.

cons:
1. The paper introduces two submethods, utility modulation and focus modulation, which appear to be somewhat disjointed. The integration of these two methods into a single framework seems somewhat forced, and their connection could be more clearly articulated to enhance the coherence of the overall approach.
2. The utility modulation technique, which involves manipulating model outputs, may be considered too straightforward and potentially trivial. The concept of introducing noise to influence model performance is a well-known principle, and the paper might not sufficiently highlight its novelty or the specific advantages of this approach over existing methods.
3. The focus modulation section of the paper is overly simplified and lacks in-depth explanation. The methodology for directing model logits to concentrate on specific knowledge areas is not adequately detailed, leaving readers unsure about the practical application and the intricacies involved in implementing this technique.

**Questions:**

1. Figure 1 lacks useful information. I cannot understand the meaning of logits redistribution expressed in it. would the author explain it in detail?
2. Theorem 3 does not seem to explain why the model can enhance the performance of some classes. Could the author further analyze it?
3. Could the authors provide a detailed explanation of how the focus modulation is trained?
4. Additionally, it would be beneficial to understand how this technique can be applied to the LLAMA3 model. Is focus modulation exclusively suitable for semantic segmentation tasks, or does it have broader applicability?
5. The addition of noise to features seems conceptually similar to the use of soft prompts. Could the authors provide an analysis comparing their method with soft prompts in terms of effectiveness and versatility?

**Reviewer Confidence:**

3: The reviewer is confident but not certain that the evaluation is correct

**Scope:**

3: The work is somewhat relevant to the Web and to the track, and is of narrow interest to a sub-community

---

### Official Review · Reviewer_bug7 · 2024-12-01

**Novelty:** 5
**Technical Quality:** 4

**Review:**

This is an interesting work. The authors propose a method that achieves task adaptation for pre-trained models via logits redistribution, enabling the models to match additional tasks without re-training. They provide theoretical analysis of their framework and validate its performance on various datasets and models.
The research problem addressed in this paper is highly valuable, but the proposed solution has certain limitations. Specifically, the paper lacks guidance on how to set $\sigma$ for a given user or task and does not evaluate the runtime or computational overhead of the modulated models.

**Questions:**

1. The authors claim that Aim enables a single model to exhibit diverse behaviors \textbf{meeting the specific needs of stakeholders}, but didn't show how the specific needs can be connected to the parameter $\sigma$ or other component of their paradigm. This omission significantly undermines the contributions of this work.
2. The control functions on page 3 and page 4 are represented in $\Phi$ and $\phi$ respectively, but they are likely to be the same.
3. The theoretical analysis of focus modulation is not clear enough, which doesn't clarify which parts in 3.3.1 represent the enhancing metric E.
4. The authors claim in Section 5 that they enable post-training adaptation of utility and feature prioritization. I'm confused about how to define feature prioritization in this paradigm, and how to evaluate it.

**Reviewer Confidence:**

3: The reviewer is confident but not certain that the evaluation is correct

**Scope:**

3: The work is somewhat relevant to the Web and to the track, and is of narrow interest to a sub-community

---

### Official Review · Reviewer_d2zV · 2024-12-06

**Novelty:** 5
**Technical Quality:** 5

**Review:**

The paper addresses a pressing challenge in AI model deployment: the inefficiency of maintaining multiple specialized versions of large-scale pre-trained models. The introduction of Aim as a paradigm for model modulation is novel and highly relevant to both academia and industry, and the logits redistribution strategy is an innovative mechanism to achieve modulation without retraining or additional data.

**Questions:**

1.this paper is good, but seems not quite relevant to the web. More applications and related tasks should be mentioned in this paper, to highlight its contribution for the web community.

2.While the concept of logits redistribution is introduced, the exact mechanics of how utility and focus modulation are operationalized remain somewhat vague. More details are needed to fully understand the modulation process.

**Reviewer Confidence:**

2: The reviewer is willing to defend the evaluation, but it is likely that the reviewer did not understand parts of the paper

**Scope:**

2: The connection to the Web is incidental, e.g., use of Web data or API